# Deepfake Detection through Color-Based Spatial-Temporal Feature Mapping with Biometric Information

## Abstract

The detection of deepfakes continues to grapple with challenges arising from the rapid evolution of generative models and the intricate characteristics of real-world data. Current detection frameworks frequently exhibit overfitting to particular artifacts, which constrains their effectiveness against novel manipulation techniques. While many models demonstrate high accuracy on standardized benchmark datasets, their performance often deteriorates when confronted with authentic deepfake instances. This study investigated the integration of biometric data, explicitly addressing the limitations of deepfake generation in mirroring the subtle biometric variations present in human faces. By segmenting facial regions into mesh representations, we analyzed the correlation between RGB features and biometric signals, particularly focusing on heart rate data. This approach enabled the development of Color-Based Spatial-Temporal (CST) feature maps, which provide a more nuanced depiction of the interactions between visual attributes and biometric inputs. The goal of this study was to propose a novel feature map and evaluate its performance. We assessed the effectiveness of these biosignal feature maps in conjunction with established detection models on the FaceForensics++ and Celeb-DF datasets. The incorporation of these feature maps resulted in remarkable outcomes, achieving nearly 99% accuracy (ACC) and an area under the curve (AUC) nearing 1. Importantly, our method demonstrates strong effectiveness in detecting low-quality deepfakes images with high compression level. Transitioning to a transfer learning framework, while retaining the biosignal feature maps, yielded further enhancements in performance metrics. These findings underscore the considerable value of integrating biometric information to bolster deepfake detection capabilities, often surpassing the results of prior research while remaining anchored in fundamental learning principles. The model exhibited consistent performance across diverse cross-testing scenarios, highlighting its robustness and adaptability.

## 1 Introduction

Deepfake technology leverages sophisticated artificial intelligence and deep learning techniques to generate hyper-realistic synthetic media, encompassing images, audio, and video content. Central to its functionality are neural networks, which facilitate tasks such as face swapping and voice imitation Alanazi et al. (2025). The evolution of deepfakes was catalyzed by the original use of autoencoders Kingma et al. (2013) for data compression, which were subsequently enhanced by Generative Adversarial Networks (GANs) Goodfellow et al. (2020) and Diffusion models Dhariwal & Nichol (2021), significantly elevating the realism of produced visuals and sounds Amerini et al. (2025).

Deepfakes exploit statistical patterns in authentic media, making reliable detection increasingly difficult. Their growing realism enables misuse in political manipulation, nonconsensual adult content, and disinformation, raising critical security concerns. In 2023, an estimated 500,000 deepfakes circulated on social media, underscoring the urgency of this challenge Alanazi et al. (2025); Kaur et al. (2024).

As AI-generated media blurs reality, risks of identity theft and cyber threats rise. A notable incident in 2019 showed a €220,000 fraud using AI voice cloning J. Damiani (2019). Ethical issues like consent, privacy, and misinformation remain critical.

These challenges have sparked discussions around regulatory frameworks, digital literacy enhancements, and advanced detection methodologies. The rapid proliferation of deepfakes threatens to undermine public trust in digital media, potentially leading to a phenomenon known as the 'Liar's Dividend,' wherein the veracity of authentic content is called into question, endangering social cohesion and the integrity of democratic processes Schiff et al. (2023).

Detection strategies for deepfakes can be categorized into several methodologies Alrashoud (2025):

- Visual artifact analysis detects anomalies in deepfake images and videos, while temporal methods (RNNs, LSTMs, optical flow) capture time-based inconsistencies. Key approaches include the Recurrent Convolutional Model (RCN) and Temporally Aware Pipelines. Single-frame analysis targets facial distortions and 3D head pose issues, and PRNU identifies device-specific signatures Lukas et al. (2006). Deep Features-based Methods Hsu et al. (2020); Guo et al. (2021) automate feature extraction, while shallow classifiers (e.g., SVMs, random forests) distinguish authentic from manipulated content.

- Audio deepfake detection focuses on spectral and acoustic analysis of TTS and voice cloning but struggles to generalize beyond academic datasets Schäfer et al. (2024).

- Multimodal detection integrates visual, audio, and behavioral cues—such as lip-sync inconsistencies—to enhance robustness over single-modality approaches Muppalla et al. (2023); Wang & Huang (2024).

  Biometric Signal Analysis Patil et al. (2023) examines subtle physiological cues that deepfakes often fail to mimic. Eye blinking Agrawal & Haneef (2025), eye movement Li et al. (2021), eyebrow traits Nguyen & Derakhshani (2020), and ear–mouth dynamics Gerstner & Farid (2022) reveal irregularities introduced by synthetic generation. Heartbeat detection via rPPG Hernandez-Ortega et al. (2020) further captures inconsistencies in skin-tone–based pulse signals.

In this paper Kim et al. (2025), the authors present novel forgery detection approach based on pixel-wise temporal frequency. Unlike conventional approaches that stack frequency spectra across frames, their method applies a 1D Fourier transform along the time axis at each pixel to extract features sensitive to unnatural movements.

FakeTransformer exploits pixel-level blood volume variations, enhanced with multi-scale Eulerian Video Magnification to generate MEMSTmaps from averaged YUV channels of 15 ROIs. A Vision Transformer then learns spatio-temporal descriptors for classification, achieving 98.98%, 98.50%, 98.49%, and 86.50% accuracy on DeepFakes, Face2Face, FaceSwap, and NeuralTextures, respectively, in intra-dataset evaluation Sun et al. (2022).

However, these approaches frequently struggle under high compression levels, where critical visual cues are heavily degraded and compression lowers detection accuracy, and social-media processing further masks manipulation cues Wu et al. (2022). Therefore, we propose a novel methodology for generating feature maps that embed biometric information and validate their effectiveness on public datasets. The maps incorporate the heart rate derived from facial skin and spatial and temporal information. An AI model trained on the proposed maps achieves impressive performance in high compression level of dataset FF++, demonstrating the utility of physiological features in addressing limitations of prior work.

## 2 METHODOLOGY

### 2.1 DATASET

We conducted extensive experiments on two widely-used deepfake detection benchmarks—FaceForensics++, and Celeb-DF (v1) to rigorously assess the performance and generalization ability of our proposed method across both in-dataset and cross-dataset settings.

**FaceForensics++ (FF++)** Rossler et al. (2019) provides a balanced dataset with 1000 real and 1000 fake videos, manipulated through four distinct methods: Deepfakes (DF), Face2Face (F2F), FaceSwap (FS), and NeuralTextures (NT). We utilize the heavy compression (c40) version to ensure reliable evaluation under visually consistent conditions and also compare in light compression (c23) level.

**Celeb-DF (v1) (CDFv1)** Li et al. (2020b) addresses the quality limitations of prior datasets by including 408 real and 795 fake videos generated with improved synthesis techniques. It covers 59 celebrity subjects under a wide range of facial expressions, head poses, and lighting conditions, allowing us to test robustness under realistic scenarios.

## 2.2 COLOR-BASED SPATIAL-TEMPORAL FEATURE MAP GENERATION

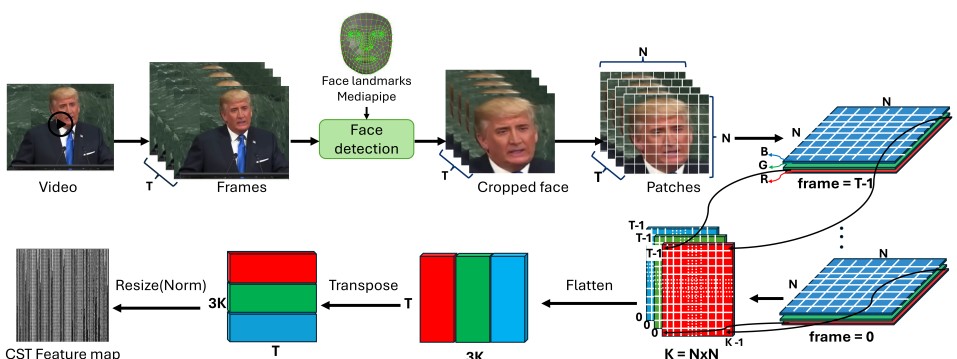

Figure 1: A Color-based Spatial-Temporal (CST) feature map embedding process.

Video analysis detection methods are typically grouped into spatial (frame-level), temporal (sequence-level), and multimodal (cross-modal) approaches. Some address spatial biases within frames, while others mitigate temporal artifacts across sequences Ramanaharan et al. (2025). Recent advances often employ transformers and hybrid architectures. We propose a novel Color-based Spatial-Temporal (CST) feature map, derived from the RGB values of facial images, to capture physiological signals correlated with biometric data. Remote heart rate estimation relies on detecting subtle color variations in facial skin over time, which reflect blood volume changes and can be used to extract remote photoplethysmography (rPPG) signals Xiao et al. (2024). Our CST feature map encodes these color fluctuations, effectively representing the rPPG component and enabling accurate non-contact heart rate measurement, a mechanism that has been validated in prior studies Verkruysse et al. (2008); Xiao et al. (2024); Hassan et al. (2017); Liu et al. (2024).

The embedding process for the CST feature map is depicted in Figure 1. It begins by extracting frames from video content using the Mediapipe face landmark detector, which identifies key facial landmarks. A total of 468 keypoints are captured, allowing us to crop images around the facial region. These cropped images are resized to dimensions $H \times W$ before being divided into $N \times N$ patches. For this study, the value of $N$ is held constant, and its effect on performance is not the focus here. Each patch is generated from the facial frame, where each video frame $F_t \in \mathbb{R}^{H \times W \times 3}$ is segmented into uniform $N \times N$ non-overlapping patches. The patches represent sub-regions of size $\frac{H}{N} \times \frac{W}{N}$ located at specific grid coordinates (i,j), with $i, j \in \{0, 1, \ldots, N-1\}$.

For each patch $P_k^t$, defined by the flattened index $k = i.N + j$, the mean RGB values are computed using the formula:

$$s_k^t = \frac{1}{|P_k^t|} \sum_{(x,y) \in P_k^t} F_t(x, y), \tag{1}$$

where $s_k^t \in \mathbb{R}^3$ represents the average RGB vector for patch $k$ at frame $t$, and $F_t(x, y) \in \mathbb{R}^3$ denotes the RGB values at the pixel coordinates $(x, y)$ within the frame $F_t$.

By iterating this process through a sequence of $T$ frames, a 3D tensor is constructed:

$$\mathbf{S} \in \mathbb{R}^{K \times T \times 3}, \tag{2}$$

where $K = N^2$ denotes the total number of patches. Each tensor entry, $\mathbf{S}_{k,t,:}$, captures the RGB signal for patch $k$ at frame $t$.

The CST map is then formulated by flattening the patch and channel dimensions, followed by transposing the tensor:

$$\mathbf{M} = \text{reshape}(\mathbf{S}, (T, 3K))^\top \in \mathbb{R}^{3K \times T}. \tag{3}$$

This resulting matrix $\mathbf{M}$ effectively encapsulates the temporal progression of color values across each patch-channel combination.

For practical applications, we normalize $\mathbf{M}$ to a range of $[0, 255]$ and resize it to a standard resolution (e.g., $224 \times 224$) using bilinear interpolation:

$$\mathbf{M}_{\text{norm}} = \text{resize}\left(\text{norm}_{[0,255]}(\mathbf{M})\right). \tag{4}$$

This CST map explicitly encodes temporal changes of RGB values across frames, often including spatial aggregation. It transforms the raw RGB sequences into a structured representation where subtle color fluctuations corresponding to blood volume changes are highlighted, making it easier for a model to learn rPPG-related patterns and suitable for diverse applications, including physiological signal analysis and deepfake detection.

Deepfake detection methods that utilize raw RGB signals can be classified into three main categories: techniques focused on pixel-level inconsistencies and artifacts, those analyzing the statistical characteristics of texture and color channels, and methods investigating the frequency domain of raw RGB data. The approach presented here is unique in that it simultaneously evaluates both spatial and temporal attributes of the RGB signals, delivering a more thorough analysis.

Deepfake detection methodologies are increasingly using biometric signals, particularly remote photoplethysmography (rPPG) from RGB video. These methods assume that heart rates from genuine and manipulated faces differ significantly. The DeepFakesON-Phys Hernandez-Ortega et al. (2020) framework employs a convolutional attention network (CAN) to extract features, achieving over 98% AUC on datasets like Celeb-DF v2. However, a recent study (2025) challenges the idea that deepfake generation compromises rPPG signals, showing that high-quality deepfakes can produce authentic heart rates. This suggests that pulse detection alone is insufficient, highlighting the need for new detection strategies that analyze blood flow patterns and the importance of advanced rPPG techniques in identifying subtle abnormalities in biometric data. Thus, the trial in this study is to evaluate comprehensive raw RGB signals, including all biometric information.

## 2.3 MODEL ARCHITECTURES

In this study, we evaluate our proposed method utilizing the Xception architecture Chollet (2017), which is a lightweight yet powerful convolutional neural network that incorporates depthwise separable convolutions. Each separable convolution operation consists of a depthwise convolution—performing spatial convolutions independently across each channel—followed by a 1×1 pointwise convolution that captures inter-channel correlations. This architecture is predicated on the premise that spatial and cross-channel feature dependencies can be disentangled and learned sequentially as a 2D mapping followed by a 1D mapping, rather than being handled jointly in a 3D context.

The Xception model consists of 36 convolutional layers grouped into 14 modules, with residual connections implemented in all modules except the first and the last. A softmax output layer is employed to facilitate classification tasks. For our evaluations, both intra-dataset comparisons and cross-manipulation assessments are conducted using the standard Xception model ($batchsize = 32, learningrate = 0.001$) to contrast performance variations between frame-level datasets and color-based spatial-temporal (CST) feature map datasets.

In our cross-dataset evaluation, we employ child Xception model on dataset without CST with pretrained weights from a parent Xception model, which was previously trained on a CST feature map

| Dataset | (w/o) CST (videos) | (w/) CST (images) |
|---------|--------------------|--------------------|
| FF++ | Real: 1000
DF: 1000
F2F: 1000
FS: 1000
NT: 1000 | Real: 1000
DF: 1000
F2F: 1000
FS: 1000
NT: 1000 |
| CDFv1 | Real: 408
Fake: 795 | Real: 408
Fake: 795 |

Table 1: The details of the evaluation datasets, including the dataset without(w/o) CST feature map dataset and with(w/) CST feature map.

for deepfake detection. We specifically load the parent model's checkpoint and selectively transfer only those layers that have matching shapes with the child model. The classifier head is reconfigured to output two classes (real and fake), enabling fine-tuning on the frame dataset. This strategy capitalizes on the rich feature representations acquired by the parent model, thereby enhancing convergence speed and performance in the target domain.

## 2.4 FUNCTION LOSS

In our approach, we utilize cross-entropy loss as the objective function to enhance classification performance. This loss function is a standard choice for both binary and multi-class classification tasks, as it quantifies the divergence between the predicted probability distributions and the actual labels. For binary classification, the cross-entropy loss is expressed as follows:

$$\mathcal{L}_{CE} = -\left[ y \log(\hat{y}) + (1-y) \log(1-\hat{y}) \right], \tag{5}$$

where $y$ denotes the actual label of the sample (0 or 1) and $\hat{y}$ represents the predicted probability of the positive class (class 1). Extending to multi-class classification with $C$ possible classes, the loss is defined as:

$$\mathcal{L}_{CE} = -\sum_{i=1}^{C} y_i \log(\hat{y}_i). \tag{6}$$

Here, $y_i$ corresponds to the actual probability for class $i$, typically encoded as 1 for the correct class and 0 for all others. At the same time, $\hat{y}_i$ indicates the predicted probability for class $i$. This formulation effectively captures the model's predictive accuracy across multiple classes.

## 3 EXPERIMENTS

### 3.1 EXPERIMENT SETTINGS

All experiments were rigorously performed on an NVIDIA GeForce RTX 4090 GPU, utilizing driver version 550.54.14 and equipped with 24GB of memory. The deep learning framework employed was PyTorch 2.1.1, with CUDA 11.8 support, executed within a Python 3.8.18 environment.

### 3.2 EXPERIMENT RESULTS

#### 3.2.1 CST FEATURE MAP DATASETS:

Table 1 outlines the evaluation datasets, which encompass dataset without(w/o) CST feature map and the proposed CST feature map representations derived from FF++ and CDFv1. The dataset without(w/o) CST feature map is composed of isolated video frames, whereas the CST feature map dataset integrates our CST embedding methodology. In Figure 2, an example from the FF++ dataset is showcased, displaying both the dataset without(w/o) CST feature map and its associated CST feature map. Notably, the CST feature maps exhibit distinctive patterns that effectively differentiate authentic videos from those altered by a range of manipulation techniques.

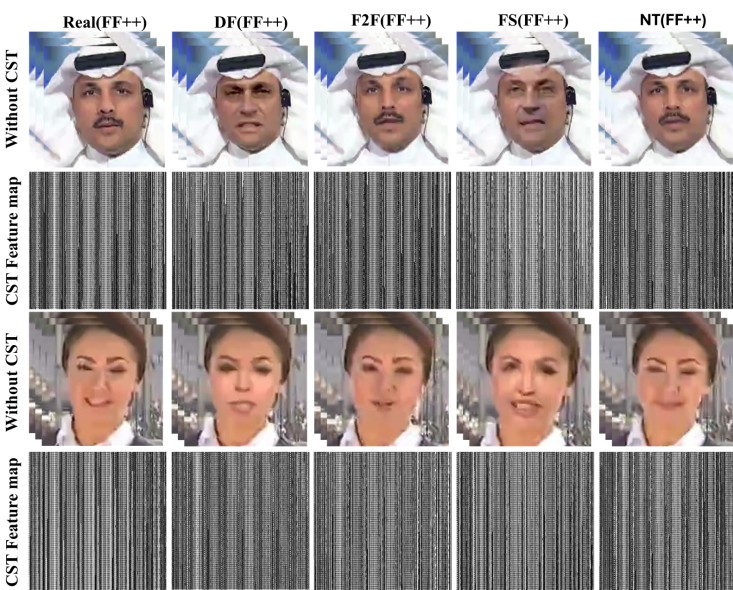

Figure 2: The example of dataset without(w/o) CST feature map and the corresponding dataset with(w/) CST feature map of the video on FF++ dataset.

| Dataset | Metrics | Deepfakes | Face2Face | FaceSwap | NeuralTextures | FF++ | CDFv1 |
|---------|---------|-----------|-----------|----------|----------------|------|-------|
| (w/o) CST | ACC | 93.94 | 85.12 | 88.78 | 59.31 | 77.20 | 94.41 |
|  | AUC | 98.66 | 93.50 | 95.54 | 95.49 | 75.07 | 99.58 |
| (w/) CST | ACC | **99.75** | **98.13** | **98.53** | **98.62** | **99.9** | **100** |
|  | AUC | **100** | **99.86** | **99.93** | **99.91** | **100** | **100** |

Table 2: The results of intra-dataset evaluation between the dataset without(w/o) CST feature map and the with(w/) CST feature map on Xception model.

### 3.2.2 INTRA-DATASET EVALUATION

Table 2 summarizes the results of the intra-data set evaluation, contrasting the performance of the dataset without CST with that of the CST feature map dataset. The bold font indicates the highest AUC and ACC. In particular, the Xception model exhibits a marked improvement when applied to the CST feature map dataset, consistently achieving accuracy scores close to 100%, significantly surpassing its performance on the dataset without CST. The proposed method boosts accuracy from 75.07% to 100% on FF++ (c40) and from 94.41% to 100% on CDFv1 using CST maps. Overfitting does not occur, as the model performs well on both the training and validation sets. Furthermore, cross-manipulation and cross-dataset evaluations are presented in the following sections to further demonstrate this.

In Table 3, we detail the accuracy (%) results for dataset with CST and without CST feature map datasets across various architectures, including Xception, ResNet50 He et al. (2016), EfficientNet Tan & Le (2019), and ViT Dosovitskiy et al. (2020). The best ACC scores are highlighted in bold. The models were trained on FF++ (c40), encompassing five classes: "real," "deepfakes," "face2face," "faceswap," and "neuraltextures". The CST feature map representation shows a considerable enhancement in performance for CNN-based models relative to the dataset without CST, registering accuracy increases of +26.45% for Xception, +21.00% for ResNet50, and +23.93% for EfficientNet. However, these gains are not reflected in the ViT-based model, which shows a performance decrease of 7. 71% compared to the dataset without CST.

To further underscore the efficacy of the CST feature map, we present t-SNE visualizations Maaten & Hinton (2008) in Figure 3. These visualizations depict the feature distributions of the Xception, ResNet50, EfficientNet, and ViT models in both datasets. Each color corresponds to one of the five classes: "real," "deepfakes," "face2face," "faceswap," and "neuraltextures." In the dataset with

| Dataset | Xception | ResNet50 | EffNet | ViT |
|---|---|---|---|---|
| (w/o) CST | 73.05 | 72.27 | 71.57 | **44.23** |
| (w/) CST | **99.5** | **93.27** | **95.50** | 36.52 |

Table 3: The ACC (%) results of the without(w/o) CST dataset and with(w/) CST feature map dataset on Xception, ResNet50, EfficientNet (EffNet), and ViT. The detectors are trained on FF++ (c40) with 5 classes: "real", "deepfakes", "face2face", "faceswap", "neuraltextures".

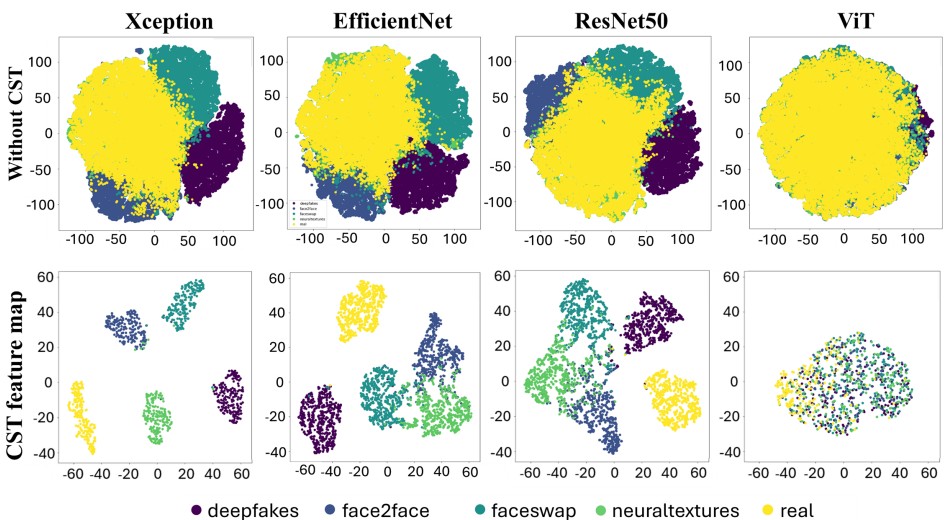

Figure 3: The t-SNE visualization on intra-dataset scenario of Xception, EfficientNet, ResNet50, ViT. The detector is trained on FF++ (c40), and different colors represent different subsets (corresponding to 5 classes).

CST, the CNN models exhibit moderate separation between forgery types, with significant overlap between authentic and manipulated images, which poses a heightened risk of misclassification. Conversely, the CST feature map representation facilitates more apparent inter-class separation and tighter intra-class clustering, particularly among distinct forgery types. This indicates that the CST representation not only enhances the model's capacity to differentiate real from fake but also fosters improved clustering of samples from the same manipulation class. The results suggest that integrating such discriminative representations could further elevate the generalisation capabilities of deepfake detectors by leveraging manipulation-specific features. The t-SNE results reveal poor class clustering for ViT, reflecting weak discriminative features. ViT emphasises global patterns, but lacks the local feature sensitivity that CNNs provide.

### 3.2.3 CROSS-MANIPULATION EVALUATION

Table 4 reports cross-manipulation results for the dataset without CST and with CST feature map, measured by accuracy ACC (%) and AUC (%). The best scores are highlighted in bold. In first four settings, models are trained on one manipulation type and tested on others; in the fifth, one type is held out for validation and the remaining three type is for training. Across all cases, with CST achieves ACC and AUC above 90%, approaching 100% in some, and consistently outperforms without CST dataset by factors of 2–3 times, demonstrating the strength of the proposed representation.

### 3.2.4 COMPARISION IN DIFFERENT COMPRESSION LEVELS

Table 5 compares the AUC (%) performance of our CST feature map (both c23 and c40) representation with existing methods under cross-manipulation and cross-dataset evaluations. The best and second-best results are marked in bold and underlined, respectively. In cross-manipulation, CST in c40 level attains an average AUC of 99.97%, outperforming all baselines, including a 0.43% gain over CST on c23 level and a 2.32% gain over UCF. Figure 4 further illustrates this strength: (a) con-

| Train set | Data Type | Metrics | DF | F2F | FS | NT |
|---|---|---|---|---|---|---|
| DF | (w/o) CST | ACC | – | 50.93 | 48.77 | 47.87 |
| | | AUC | – | 52.89 | 52.87 | 49.84 |
| | (w/) CST | ACC | – | **89.25** | **97.17** | **97.25** |
| | | AUC | – | **98.26** | **99.75** | **99.78** |
| F2F | (w/o) CST | ACC | 52.47 | – | 47.72 | 46.73 |
| | | AUC | 58.56 | – | 46.95 | 39.72 |
| | (w/) CST | ACC | **100** | – | **100** | **99.75** |
| | | AUC | **100** | – | **100** | **100** |
| FS | (w/o) CST | ACC | 59.43 | 48.27 | – | 47.28 |
| | | AUC | 66.10 | 41.34 | – | 36.16 |
| | (w/) CST | ACC | **99.75** | **99.75** | – | **100** |
| | | AUC | **100** | **99.94** | – | **100** |
| NT | (w/o) CST | ACC | 46.69 | 42.39 | 39.60 | – |
| | | AUC | 46.32 | 33.75 | 30.58 | – |
| | (w/) CST | ACC | **100** | **99.75** | **100** | – |
| | | AUC | **100** | **99.94** | **100** | – |
| FF++ | (w/o) CST | ACC | 71.86 | 58.51 | 55.41 | 33.20 |
| | | AUC | 60.53 | 45.66 | 50.01 | 40.06 |
| | (w/) CST | ACC | **99.83** | **99.67** | **99.75** | **99.58** |
| | | AUC | **100** | **99.99** | **99.98** | **99.92** |

Table 4: The results of cross-manipulation evaluations between the dataset without(w/o) CST and the with(w/) CST feature map in ACC(%) and AUC(%) in FF++ (c40) dataset (Deepfakes (DF), FaceSwap (FS), Face2Face (F2F), and NeuralTextures (NT)) on Xception model.

| Model | Reference | Cross-manipulation | | | | | Cross-dataset |
|---|---|---|---|---|---|---|---|
| | | DF | F2F | FS | NT | Avg. | CDFv1 |
| Xception | CVPR2017 | 60.53 | 45.66 | 50.01 | 40.06 | 49.06 | 77.43 |
| EfficientB4 Tan & Le (2019) | PMLR2019 | 97.57 | 97.58 | 97.97 | 93.08 | 96.55 | 79.09 |
| CNN-Aug Wang et al. (2020) | CVPR2020 | 90.48 | 87.88 | 90.26 | 73.13 | 85.44 | 74.20 |
| X-ray Li et al. (2020a) | CVPR2020 | 97.94 | 98.72 | 98.71 | 92.90 | 97.06 | 70.93 |
| SPSL Liu et al. (2021) | CVPR2021 | 97.81 | 97.54 | 98.29 | 92.99 | 96.66 | **81.50** |
| SRM Luo et al. (2021) | CVPR2021 | 97.33 | 96.96 | 97.40 | 92.95 | 96.16 | 79.26 |
| CORE Ni et al. (2022) | CVPRW2022 | 97.87 | 98.03 | 98.23 | 93.39 | 96.88 | 77.98 |
| Recce Cao et al. (2022) | CVPR2022 | 97.97 | 97.79 | 97.85 | 93.57 | 96.79 | 76.80 |
| UCF Yan et al. (2023) | ICCV2023 | 98.83 | 98.40 | 98.96 | 94.41 | 97.65 | 77.90 |
| TCAN Amin et al. (2024) | ERA(2024) | 99.16 | 98.93 | 99.03 | 98.15 | 98.82 | – |
| RAE Tian et al. (2024) | ECCV2024 | 99.60 | 99.10 | 99.20 | 97.60 | 98.90 | – |
| FreqBlender Hanzhe et al. (2024) | NeurIP2024 | 99.18 | 96.76 | 97.68 | 90.88 | 96.13 | – |
| BSF Kim et al. (2025) | ICCV2025 | 99.90 | 97.10 | 99.80 | 96.90 | 98.42 | – |
| Ours-Xception trained with CST | – | 98.22 | 99.92 | **100** | **100** | 99.54 | 58.59 |
| Ours-Xception trained with CST (c40) | – | **100** | **99.99** | 99.98 | 99.92 | **99.97** | 80.34 |

Table 5: AUC(%) on the cross-manipulation and cross-dataset evaluation. Deepfakes (DF), FaceSwap (FS), Face2Face (F2F), and NeuralTextures (NT). The models are trained on FF++(c23). The 'Avg.' column denotes the mean AUC computed over various datasets. The final line is the result of our CST evaluated on FF++(c40).

fusion matrices show only 1–3 errors per class, and (b) t-SNE embeddings reveal clearly separated trajectories for real (yellow) and fake (purple) samples.

For cross-dataset evaluation, CST on c40 improves AUC by 2.92% over without(w/o) dataset baseline and ranks second overall, just 1.16% behind SPSL. Figure 5 highlights this advantage with GradCAM visualizations, where Xception misclassifies a without CST sample (red box), but CST yields correct predictions.

# 4 CONCLUSIONS

This study presented a novel methodology for generating color-based spatio-temporal (CST) feature maps by analyzing RGB facial features and heart rate data. The integration of these feature maps into existing models has led to impressive results:

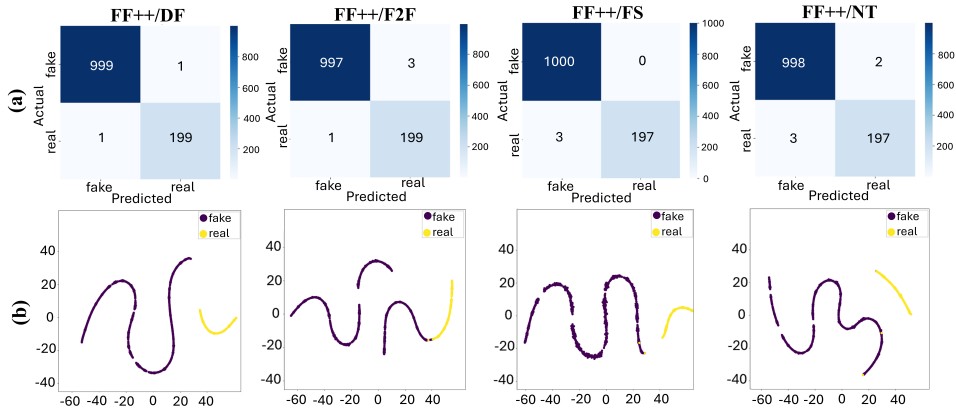

Figure 4: (a) Confusion matrix and (b) t-SNE visualization from cross-manipulation evaluation using the CST feature map. Trained on three manipulations, tested on the remaining one (FF++ c40).

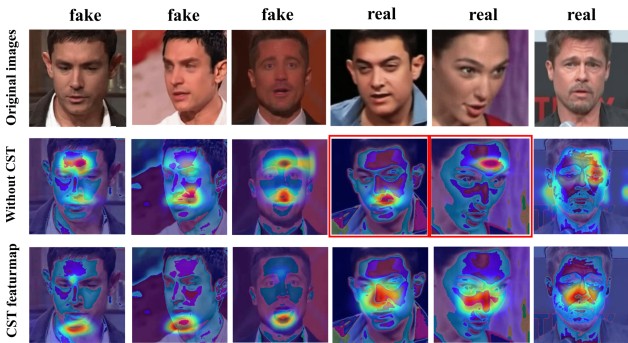

Figure 5: GradCAM visualizations on some samples with CST feature map dataset and without CST dataset in cross-dataset evaluation. Red rectangle marks misclassification.

- Intra-dataset evaluation: Using CST feature maps with the Xception architecture, model achieved nearly 100% accuracy and an AUC close to 1, significantly improving from 77.20% accuracy without CST data to 99.9% with CST maps in full FF++ (c40) dataset. And achieving an improvement from 94.41% to 100% on the CDFv1 dataset.

- Cross-manipulation evaluation: CST feature maps showcased strong performance in heavy compression across various manipulation types (DF, F2F, FS, NT), achieving over 90% accuracy and AUC values, with averages reaching 99.97%, greatly surpassing baseline models.

The key contributions are follows:

- Enhanced Generalization: CST feature maps improve model adaptability to new deepfake types, reducing overfitting issues common in previous research.

- Raw RGB Data Utilization: The method captures discrepancies directly from raw RGB signals, identifying forgery indicators that processed images might miss.

- Practical Model Feasibility: Transfer learning enhances performance, offering a solution to challenges associated with data scarcity in deepfake detection with heavy compression media.

In conclusion, this research significantly advances deepfake detection technology, demonstrating that CST feature maps combined with biometric data improve generalization and robustness in real-world applications, laying a foundation for future detection strategies.

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
