**Supplementary Material for ICLR Submission #8450 Discussion**

1. Comparison of different color spaces

| Color spaces | Metrics | FF++/DF | FF++/F2F | FF++/FS | FF++/NT |
|---|---|---|---|---|---|
| RGB | ACC | **99.83** | **99.67** | **99.75** | **99.58** |
| | AUC | **100** | **99.99** | **99.98** | **99.92** |
| YUV | ACC | 75.25 | 63.25 | 92.00 | 67.00 |
| | AUC | 91.75 | 99.85 | 99.92 | 91.64 |
| YCbCr | ACC | 50.75 | 50.00 | 50.00 | 50.00 |
| | AUC | 44.79 | 60.42 | 59.25 | 74.97 |

Table 1: Comparison in accuracy (ACC %) and Area Under the Curve (AUC %) of different color spaces (RGB, YUV, YCbCr) in FF++ dataset (c40).

Table 1 presents the comparison results in terms of accuracy (ACC, %) and AUC (%) across different color spaces (RGB, YUV, YCbCr) on the FF++ (c40) dataset under cross-manipulation evaluation. RGB achieves the best performance, reaching nearly 100% in all metrics. Because different channels of the RGB signal had varying relative strengths of PPG signals. In RGB color space, the green channel delivers the highest signal quality. The green channel contains the strongest pulsatile signal of PPG [1]. Because hemoglobin is most sensitive to changes in oxygenation of green light absorption. According to the formula of converting from RGB to YUV and RGB to YCbCr, the ratio of green channel is reduced, and green channel is diluted when switched to YcbCr, YUV.

$$
\begin{bmatrix} Y \\ Cb \\ Cr \end{bmatrix} = \begin{bmatrix} 0.299 & 0.587 & 0.114 \\ -0.168736 & -0.331264 & 0.5 \\ 0.5 & -0.418688 & -0.081312 \end{bmatrix} \cdot \begin{bmatrix} R \\ G \\ B \end{bmatrix} + \begin{bmatrix} 0 \\ 128 \\ 128 \end{bmatrix}
$$

$$
\begin{bmatrix} Y \\ U \\ V \end{bmatrix} = \begin{bmatrix} 0.299 & 0.587 & 0.114 \\ -0.14713 & -0.28886 & 0.436 \\ 0.615 & -0.51499 & -0.10001 \end{bmatrix} \cdot \begin{bmatrix} R \\ G \\ B \end{bmatrix}
$$

2. Comparison of different compression levels in FF++ dataset

Overall, table 2 shows that the performance at compression level C40 surpasses that of C23, with improvements of 5.11% in ACC and 0.43% in AUC, respectively. The C40 setting introduces stronger compression artifacts—such as block noise, color distortions, and temporal instability—which amplify the discrepancy between real and fake content, thereby making deepfakes easier to detect than under C23. Our color–spatial–temporal feature map is inherently sensitive to these compression-induced distortions, resulting in naturally higher performance at C40. Consequently, our method demonstrates strong effectiveness in low-quality deepfake detection. Furthermore, the cross-manipulation evaluation on C40 achieves the best results among all FF++ benchmarks.

| Compression level with CST | Metrics | FF++/DF | FF++/F2F | FF++/FS | FF++/NT | AVG | FF++ |
|---|---|---|---|---|---|---|---|
| c23 | ACC | 81.17 | 97.50 | **99.83** | **99.92** | 94.60 | 99.5 |
|  | AUC | 98.22 | 99.92 | **100** | **100** | 99.54 | 100 |
| c40 | ACC | **99.83** | **99.67** | 99.75 | 99.58 | **99.71** | **99.9** |
|  | AUC | **100** | **99.99** | 99.98 | 99.92 | **99.97** | **100** |

Table 2: Comparison in ACC(%) and AUC(%) in different compression levels (c23 and c40) on FF++ dataset.

| Compression level with CST | Xception | Resnet50 | Efficientnet | ViT |
|---|---|---|---|---|
| c23 | 87.3 | 83.20 | 88.4 | **44.64** |
| c40 | **99.5** | **93.27** | **95.50** | 36.52 |

Table 3: The ACC (%) results of the CST dataset with c23 compression level and c40 compression level dataset on Xception, ResNet50, EfficientNet (EffNet), and ViT. The detectors are trained on FF++ with 5 classes: "real", "deepfakes", "face2face", "faceswap", "neuraltextures".

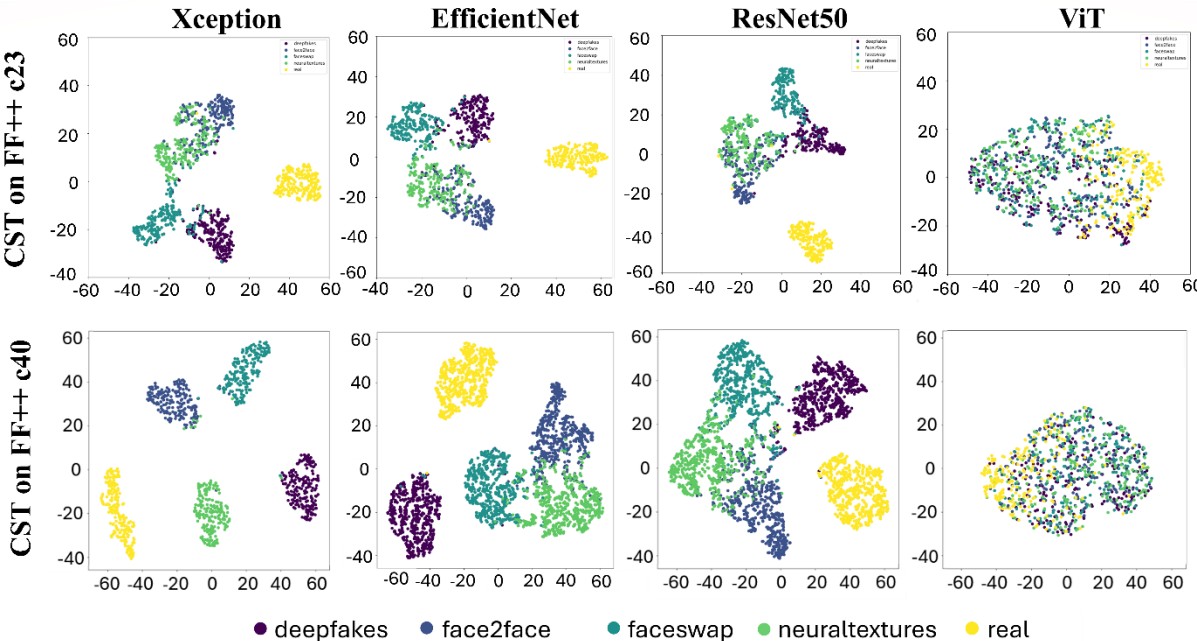

Figure 1: The t-SNE visualization on intra-dataset scenario of Xception, EfficientNet, ResNet50, ViT. The detector is trained on FF++ c40 and c23, and different colors represent different subsets (corresponding to 5 classes).

Table 3 and Figure 1 present the performance comparison of Xception, EfficientNet, ResNet-50, and ViT trained on FF++ C40 and FF++ C23. Figure 1 further provides the t-SNE visualizations of the feature embeddings obtained by these models on both compression levels. Together with the results in Table 3 and Figure 3 of the main paper,

these findings indicate that CST features exhibit strong sensitivity to the choice of backbone architecture.

CST is a 2D image-like matrix in which rows correspond to flattened patch-channel features and columns represent temporal frames, thereby embedding both local temporal and spatial correlations. While ViTs are known to excel at capturing global relationships through self-attention, they are less effective at modeling fine-grained local patterns. This limitation becomes more pronounced because CST maps flatten the spatial layout, removing the original 2D structure that ViTs typically rely on via positional embeddings. Consequently, the positional and semantic organization required by the attention mechanism is disrupted.

In contrast, CNN-based models can naturally exploit the CST representation: convolutional kernels capture local correlations along both rows (patch-channel interactions) and columns (temporal dynamics). This enables CNNs to model CST features effectively, which explains their consistently superior performance over ViTs in this setting.

3. Comparison on C40 compression level

|  | FF++/DF | FF++/FS | FF++/F2F | FF++/NT | Avg |
|---|---|---|---|---|---|
| SupCon [2] | 92.69 | 53.52 | 71.81 | 95.93 | 96.18 |
| SFFF [3] | 94.15 | 92.37 | 90.13 | 99.56 | 99.75 |
| Xception (w/) CST | **100** | **99.98** | **99.99** | **99.92** | **99.97** |

Table 4: The AUC (%) results for heavy compression level on FF++ dataset.

Table 4 presents the AUC (%) comparison under heavy compression on the FF++ dataset. Our method achieves the best cross-manipulation performance, with AUCs approaching 100%.