# OpenReview forum: "Deepfake Detection through Color-Based Spatial-Temporal Feature Mapping with Biometric Information"
_ICLR.cc/2026/Conference — Submitted to ICLR 2026_

### Official Review · Reviewer_EFWB · 2025-10-27

**Soundness:** 2
**Presentation:** 3
**Contribution:** 2
**Rating:** 2
**Confidence:** 5

**Summary:**

The paper proposes a color Spatial-Temporal (CST) map for deepfake detection. Instead of classifying individual frames, the method
converts a short face video into a single image that encodes how per-patch RGB values change over time. A CNN (mainly Xception) is trained on these CST images for real/fake classification. The paper reports in-dataset and cross-manipulation results on FF++ (c23) and cross-dataset performance on Celeb-DF (v1). A transfer-learning variant initializes a frame-based model from the CST-trained network.

**Strengths:**

The paper’s main strength is its use of physiology-linked temporal color rhythms, which are expected to provide generalizable cues than frame-level artefacts. It packages these dynamics into a compact CST image, making training and inference fast with standard CNNs.

**Weaknesses:**

(1) The novelty of the paper is limited. The core idea, using heartbeat/physiology-like color changes over time to spot fake, has been
done before in rPPG-style detectors. Studies like FakeCatcher (Ciftci et al., 2019), DeepRhythm (Qi et al., ACM MM 2020) already explored such concepts earlier.

(2) The study evaluates only on FF++ (c23) and Celeb-DF v1, both relatively established/older benchmarks; there is no evidence on more recent or harder settings (e.g., DFDC, heavy compression, diffusion fakes), means the strong generalization narrative is under-supported.

(3) The approach appears architecture-dependent: CST helps CNNs but hurts ViT (Table 3), and cross-dataset results on CDFv1 do not beat the best baseline (SPSL) (Table 5). This weakens claims of broad applicability.

(4) The experimental validation skipped basic sanity checks, like trying different grid sizes, different clip lengths, various lighting conditions, strong video compression, or hiding parts of the face. So, it is hard to decipher the robustness of the proposed approach under real-world artefacts, leaving doubts whether the model is truly using heartbeat-like signals or just leaning on easy, dataset-specific quirks.

(5) The conclusion mentions analyzing heart rate data, but the method only builds CST from RGB and never extracts rPPG. Please either add an explicit analysis showing how CST tracks heart rate, or the claim should be “color dynamics correlated with physiology”.

**Questions:**

(1) Is the train-validation-test split strictly video and identity-disjoint? How is per-video vs per-frame sampling handled?

(2) It would be great to have results on broader datasets (CDF-v2, WildDeepfake, FF++ c40/raw) and under different constraints like lighting flicker, heavy compression/noise, skin-tone & makeup.

(3) Do the authors explicitly extract heart rate (or similar)? If not, provide spectral evidence or rephrase the claim.

(4) It is not clear why ViT underperform, and is the proposed approach sensitive to the base architecture? Any experiments with light temporal models, 3D CNNs, or hybrid CNN on CST?

(5) Please address the concerns mentioned in the weakness section as well.

---

> ### Author Response · Authors · 2025-12-03
> **Thank you for your comments. Please find our responses below.**
>
> 1. Yes. The dataset split is designed such that no video or identity appears in more than one split. This ensures that the model is evaluated on completely unseen subjects and video sequences.
> Per-video sampling: a video is assigned entirely to train, val, or test.
> Per-frame sampling: frames are sampled from each assigned video; frames are never mixed across splits.
> 2. Thank you for your valuable comment. After re-examining our experiments, we confirmed that the previously reported results were obtained using the C40 compression level. This was due to an oversight during dataset preparation, and we have now corrected this in the paper. We additionally evaluated our method on the C23 level, and the updated comparison is provided in Table 2 of the supplementary material.
> Regarding the raw (uncompressed) level, the dataset size is extremely large, which makes full-scale processing infeasible within our current computational constraints. We also conducted experiments across different color spaces, as color information is an important component of our feature map. The corresponding results are presented in Table 1 of the supplementary material.
> For the evaluation of other datasets, we plan to include these in our future work. In this we will focus on the low-quality datasets and high compression level.
> The videos have different lengths. The number of frames in videos are different; it is not fixed.
> 3. The CST (Color–Spatial–Temporal) feature map explicitly encodes temporal changes of RGB values across frames, often incorporating spatial aggregation. By transforming raw RGB sequences into a structured representation, it highlights subtle color fluctuations induced by blood volume changes beneath the skin, making rPPG-related patterns more accessible for model learning. In summary, the CST feature map captures physiological signals—specifically the remote photoplethysmography (rPPG) component—through these color variations, a mechanism that has been validated in prior studies. I updated in 2.2 COLOR-BASED SPATIAL-TEMPORAL FEATURE MAP GENERATION.
> 4. CST is a 2D image-like matrix: rows = flattened patch-channel features, columns = temporal frames so local temporal and spatial correlations are embedded. It is well known that the ViTs excel at global features due to their self-attention mechanism, they have limitations with local features, while CNNs handle local feature extraction efficiently.  CST maps flatten the spatial layout; the original 2D spatial structure is lost. So ViTs fail to exploit the CST map’s structure effectively, because the positional and semantic structure expected by the attention mechanism is disrupted. In the meantime, the convolutional kernels can learn local correlations along rows and columns, capturing patterns across patch channels (rows) and temporal frames (columns). Hence, CNNs can handle CST well.

---

### Official Review · Reviewer_pRDX · 2025-10-31

**Soundness:** 3
**Presentation:** 2
**Contribution:** 2
**Rating:** 2
**Confidence:** 5

**Summary:**

The paper introduces a deepfake detection approach that integrates biometric information such as heart rate signals extracted from facial color variations, into a new feature representation called CST feature map.

**Strengths:**

1. The paper is well-written
2. The performance of intra-data set FF++ is impressive

**Weaknesses:**

1. The paper doesn't properly do cross-dataset evaluation. The results on CDFv2, DFDC, DFDCP, KoDF, DF-Platter. The current results shown doesn't indicate that the model is generalizable and will work across a variety of datasets.
2. The authors don't compare their performance of several recent methods.
CVPR 24: https://openaccess.thecvf.com/content/CVPR2024/papers/Nguyen_LAA-Net_Localized_Artifact_Attention_Network_for_Quality-Agnostic_and_Generalizable_Deepfake_CVPR_2024_paper.pdf
ECCV 24: https://www.ecva.net/papers/eccv_2024/papers_ECCV/papers/06913.pdf
NeurIPS24: https://openreview.net/pdf?id=otZPBS0un6
AAAI 25: https://arxiv.org/pdf/2501.04376

3. It is not clear to as to how CST feature maps can capture physiological features and heart rate. Can you justify ?

**Questions:**

I have mentioned the questions in the weakness section.

---

> ### Author Response · Authors · 2025-12-03
> **Thank you for your comments. Please find our responses below.**
>
> 1. Thank you for your suggestions. In future work, we plan to extend our evaluation to additional datasets. In this paper, we focus on low-quality datasets and datasets with high compression levels.
> 2. We added your suggested papers to our paper. However, in our paper, we evaluated on CDFv1, so we did not insert the results of the cross-validation for your suggested papers which use CDFv2.
> 3. I updated in part 2.2 COLOR-BASED SPATIAL-TEMPORAL FEATURE MAP GENERATION and Supplementary Material.
> The CST (Color–Spatial–Temporal) feature map explicitly encodes temporal changes of RGB values across frames, often incorporating spatial aggregation. By transforming raw RGB sequences into a structured representation, it highlights subtle color fluctuations induced by blood volume changes beneath the skin, making rPPG-related patterns more accessible for model learning. In summary, the CST feature map captures physiological signals—specifically the remote photoplethysmography (rPPG) component—through these color variations, a mechanism that has been validated in prior studies

---

### Official Review · Reviewer_9fi1 · 2025-10-31

**Soundness:** 3
**Presentation:** 4
**Contribution:** 3
**Rating:** 4
**Confidence:** 3

**Summary:**

This study enhances deepfake detection by integrating biometric signals, especially heart rate, with facial RGB features to create Color-Based Spatial-Temporal (CST) feature maps. Tested on the FaceForensics++ and Celeb-DF datasets, the approach achieved nearly 99% accuracy and demonstrated strong robustness, outperforming traditional models and showing superior adaptability through transfer learning.

**Strengths:**

1. The paper presents an innovative Color-Based Spatial-Temporal (CST) feature map combining RGB and biometric signals to capture subtle physiological inconsistencies.
2. Results demonstrate near-perfect accuracy and AUC across datasets, with consistent cross-domain robustness and clear interpretability through visual analyses.

**Weaknesses:**

The paper does not evaluate the proposed method on the latest deepfake generation systems such as Sora or Veo, which limits its validation against state-of-the-art video synthesis techniques and raises questions about its real-world robustness.

**Questions:**

1. Could the authors explain why the method was not evaluated on deepfakes generated by diffusion-based models (e.g., Stable Diffusion, Sora, Veo)? Since these systems now dominate realistic video synthesis, such experiments are crucial for demonstrating generalization to current-generation forgeries.

2. The CST feature map is said to encode biometric information such as heart rate, but the paper does not show explicit quantitative or visual evidence of this relationship. Could the authors clarify how these RGB-based temporal features were verified to correspond to genuine physiological signals rather than generic color information?

---

> ### Author Response · Authors · 2025-12-03
> **Thank you for your comments. Please find our responses below**
>
> 1. Thank you for your comment and suggestion. In this work, we focus on the compression level and low-quality datasets. We have updated the paper to include comparisons across different levels of compression. Regarding deepfakes generated by diffusion-based models (e.g., Stable Diffusion, Sora, Veo), we did not include these experiments in the current version, but we plan to investigate them in future work to evaluate the generalization of our method to these current-generation forgeries.
> 2. We updated in paper: 2.2 COLOR-BASED SPATIAL-TEMPORAL FEATURE MAP GENERATION and Supplementary Material.
> We propose a novel Color-based Spatial-Temporal (CST) feature map, derived from the RGB values of facial images, to capture physiological signals correlated with biometric data. Remote heart rate estimation relies on detecting subtle color variations in facial skin over time, which reflect blood volume changes and can be used to extract remote photoplethysmography (rPPG) signals. Our CST feature map encodes these color fluctuations, effectively representing the rPPG component and enabling accurate non-contact heart rate measurement, a mechanism that has been validated in prior studies. This CST map explicitly encodes temporal changes of RGB values across frames, often including spatial aggregation. It transforms the raw RGB sequences into a structured representation where subtle color fluctuations corresponding to blood volume changes are highlighted, making it easier for a model to learn rPPG-related patterns and suitable for diverse applications, including physiological signal analysis and deepfake detection.

---

### Meta-Review · Area_Chair_8dwg · 2026-01-07

**Summary:**

This paper proposes a Color–Spatial–Temporal (CST) feature map that converts a face video clip into a 2D representation of patch-wise RGB temporal dynamics, then uses standard classifiers for deepfake detection. The motivation is that physiological-linked color rhythms (rPPG-like signals) may offer more robust cues than artifact-based detectors.

Reviewers agree the approach is easy to follow and achieves very strong in-dataset results on FaceForensics++. However, the decision is primarily driven by concerns about (i) limited novelty relative to prior rPPG/physiology-inspired deepfake detectors, (ii) insufficient evidence that CST truly captures biometric/heart-rate information (as opposed to generic color/temporal cues), and (iii) limited evaluation breadth/generalization—restricted datasets, missing modern/diffusion-era deepfakes, and missing robustness/sanity checks.

**Reviewer Concerns:**

Addressed (partially) in rebuttal: the authors clarified that data splits are identity- and video-disjoint; they corrected an oversight that prior results corresponded to C40 and added evaluation on C23 in the supplement; they expanded the description of how CST encodes temporal RGB fluctuations and provided an explanation for why CNNs perform better than ViTs on CST; and they added several recent related-work citations suggested by reviewers.

Outstanding concerns: multiple reviewers questioned novelty relative to prior physiology/rPPG-based deepfake detection (e.g., FakeCatcher/DeepRhythm), and the rebuttal does not directly resolve this. Reviewers also requested stronger evidence that CST captures biometric/heart-rate information beyond an rPPG-based rationale (e.g., explicit analysis/verification), which remains unaddressed in the provided discussion. Finally, the evaluation breadth remains limited: reviewers requested broader cross-dataset testing (e.g., DFDC/KoDF/Celeb-DF v2/WildDeepfake) and evaluation on more modern diffusion-era deepfakes (e.g., Sora/Veo), but the rebuttal largely defers these to future work. Requests for additional comparisons against recent methods and robustness/sanity checks (lighting, compression, grid/clip length sensitivity, occlusion) are also not fully resolved.

**Reviewer Scores:**

9fi1: 4 → 4

pRDX: 2 → 2

EFWB: 2 → 2

---

### Decision · Program_Chairs · 2026-01-26

Reject